# Role of Coactivator Associated Arginine Methyltransferase 1 (CARM1) in the Regulation of the Biological Function of 1,25-Dihydroxyvitamin D_3_

**DOI:** 10.3390/cells12101407

**Published:** 2023-05-17

**Authors:** Leila J. Mady, Yan Zhong, Puneet Dhawan, Sylvia Christakos

**Affiliations:** Department of Microbiology, Biochemistry and Molecular Genetics, New Jersey Medical School, Rutgers, The State University of New Jersey, Newark, NJ 07103, USA; lmady1@jh.edu (L.J.M.);

**Keywords:** CARM1, vitamin D, 1,25-dihydroxyvitamin D3, transcription

## Abstract

1,25-Dihydroxyvitamin D3 (1,25(OH)_2_D_3_), the hormonally active form of vitamin D, activates the nuclear vitamin D receptor (VDR) to mediate the transcription of target genes involved in calcium homeostasis as well as in non-classical 1,25(OH)_2_D_3_ actions. In this study, CARM1, an arginine methyltransferase, was found to mediate coactivator synergy in the presence of GRIP1 (a primary coactivator) and to cooperate with G9a, a lysine methyltransferase, in 1,25(OH)_2_D_3_ induced transcription of *Cyp24a1* (the gene involved in the metabolic inactivation of 1,25(OH)_2_D_3_). In mouse proximal renal tubule (MPCT) cells and in mouse kidney, chromatin immunoprecipitation analysis demonstrated that dimethylation of histone H3 at arginine 17, which is mediated by CARM1, occurs at *Cyp24a1* vitamin D response elements in a 1,25(OH)_2_D_3_ dependent manner. Treatment with TBBD, an inhibitor of CARM1, repressed 1,25(OH)_2_D_3_ induced *Cyp24a1* expression in MPCT cells, further suggesting that CARM1 is a significant coactivator of 1,25(OH)_2_D_3_ induction of renal *Cyp24a1* expression. CARM1 was found to act as a repressor of second messenger-mediated induction of the transcription of CYP27B1 (involved in the synthesis of 1,25(OH)_2_D_3_), supporting the role of CARM1 as a dual function coregulator. Our findings indicate a key role for CARM1 in the regulation of the biological function of 1,25(OH)_2_D_3_.

## 1. Introduction

Calcium (Ca^2+^) is a ubiquitous and versatile intracellular signaling messenger that operates at every stage of the life of a cell, from fertilization to development and differentiation and finally programmed cell death [1,2]. Extracellular Ca^2+^ also plays an indispensable role in fundamental processes such as blood clotting, epithelial sheet maintenance of tight junctions, and skeletal remodeling [3]. In order to serve its diverse physiological functions, calcium must be ingested from the diet and absorbed by the body. Vitamin D plays a principal role in the maintenance of calcium homeostasis by increasing intestinal calcium absorption [4,5]. In order to exert its biological effects, vitamin D_3_ must be converted to its active form. In the blood, vitamin D_3_ is transported by the vitamin D binding protein (DBP) to the liver, where it is hydroxylated at C-25 by one or more cytochrome P450 vitamin D 25-hydroxylases, including CYP2R1, CYP2D11, and CYP2D25, resulting in the formation of 25-hydroxyvitamin D_3_ (25(OH)D_3_), the major circulating form of vitamin D [5,6]. Patients with mutations in CYP2R1 have 25(OH)D deficiency, suggesting that CYP2R1 is the key vitamin D 25-hydorxylase [7]. 25(OH)D_3_ is subsequently transported by DBP to the kidney, where it is hydroxylated at C-1 by the cytochrome P450 monooxygenase 25-hydroxyvitamin D_3_ 1α-hydroxylase (CYP27B1; 1α(OH)ase), resulting in the hormonally active form of vitamin D, 1,25-dihydroxyvitamin D_3_ (1,25(OH)_2_D_3_) [5,6]. 25-Hydroxyvitamin D_3_ 24-hydroxylase (CYP24A1; 24(OH)ase), a mitochondrial P450 enzyme, also found in the kidney, initiates the catabolism of 1,25(OH)_2_D_3_ [5,6]. The critical role of CYP24A1 in the catabolism of 1,25(OH)_2_D_3_ in humans was noted in studies of very young children and adults with inactivating mutations in CYP24A1 who presented with hypercalcemia and hypercalciuria [8]. Recurrent kidney stones were also noted in adult patients with inactivating mutations [9,10]. The proximal tubule of the kidney has been reported to be the principal site of the synthesis of both CYP27B1 and CYP24A1 [11]. Thus, the kidney proximal tubule plays a key role in coordinating the production of 1,25(OH)_2_D_3_ needed for the maintenance of calcium homeostasis. *CYP24A1* is the gene most transcriptionally responsive to 1,25(OH)_2_D_3_ and is involved in the metabolic inactivation of 1,25(OH)_2_D_3_ as a mechanism of feedback control to maintain physiological levels of 1,25(OH)_2_D_3_ [5,6].

1,25(OH)_2_D_3_ regulates gene expression in target cells by binding to its nuclear steroid hormone receptor, the vitamin D receptor (VDR), which heterodimerizes with the retinoid X receptor (RXR) and interacts with specific DNA sequences (vitamin D response elements; VDREs) leading to activation or repression of transcription [5,6]. Changes in gene expression by 1,25(OH)_2_D_3_ and VDR are mediated by the recruitment of coregulatory complexes. These include Mediator complex, which functions in part through recruitment of RNA polymerase II and the p160 coactivators (steroid receptor coactivator 1, 2 and 3; SRC-1, SRC-2 (GRIP1) and SRC-3) that bind liganded VDR and have histone acetyltransferase (HAT) activity. Members of the p160 family can also recruit proteins as additional, secondary VDR coactivators [6]. These secondary coactivators include proteins that mediate post translational modifications of histone or non-histone proteins, including acetylation and methylation [6]. Histone methylation has been shown to cooperate with other histone modifications such as acetylation and phosphorylation to regulate chromatin structure and affect transcription [12]. Although growing evidence has indicated a major role for methylation in gene regulation, including steroid receptor-mediated transcription, little is known about the role of methylation in VDR-mediated transcription. Determining mechanisms involved in the regulation of gene expression by 1,25(OH)_2_D_3_ is essential to our understanding of how 1,25(OH)_2_D_3_ acts to maintain calcium homeostasis and the dysregulation that occurs with aging.

Protein methyltransferases have been shown to play a major role in transcriptional regulation [12,13]. Most protein methyltransferases methylate at arginine and lysine residues. Arginine methylation is catalyzed by nine protein arginine methyltransferases (PRMT1–9) [14]. CARM1 (PRMT4) belongs to the arginine methyltransferase family (PRMT type 1, which includes PRMT1–4, PRMT6 and 8) of proteins that catalyze the asymmetric N^G^N^G^ dimethylation of arginine residues. CARM1, which methylates H3 at Arg-17 preferentially as well as H3 at Arg-26 [15] and is ubiquitously expressed, was initially identified as a transcriptional regulator by its ability to be recruited by nuclear receptors and to bind to the p160 coactivator GRIP1 (SRC2) [16]. CARM1 has been shown to function as a secondary coactivator for estrogen receptor (ER), androgen receptor (AR) and thyroid hormone receptor (TR) mediated transcription [17]. The coactivator function of CARM1 has been shown to depend on the presence of the p160 coactivator and requires CARM1 methyltransferase activity for optimal nuclear receptor-mediated transactivation [16]. Recruitment of CARM1 can also result in the methylation of members of the p160 coactivator family as well as methylation of p300/CBP and MED12 [12,18]. CARM1 has also been shown to be a coregulator for transcription factors in addition to steroid receptors, including YY1 and NFkB [19], and to act as a transcriptional repressor as well as a coactivator [20,21]. In addition to arginine methyltransferases, lysine methyltransferases also play an essential role in transcription. G9a (and GLP; G9-like protein) are SET domain lysine methyltransferases that belong to the Su(var)3-9 family of proteins that methylate histone 3 at lysine 9 (H3K9) using its C terminal SET domain [22], resulting in gene silencing. Non-histone proteins have also been identified as substrates for G9a [22]. Although G9a is generally associated with transcriptional repression, it has also been shown to act as a coactivator for steroid receptors and other transcription factors [22,23]. The coactivator function of G9a for dexamethasone-induced glucocorticoid-induced transcription requires heterochromatin 1 gamma and is regulated by G9a self-methylation and phosphorylation [24]. G9a’s function as an ER coactivator is linked to methylation of ER and histone acetylation [25]. Thus, the function of the methyltransferases, either as corepressor or coactivator, depends on the cross-talk between acetylation, phosphorylation and methylation, the interaction between protein partners, as well as the promoter and enhancer specific context of individual genes.

In this investigation, we examined the role of CARM1 methyltransferase in VDR-mediated regulation of *Cyp24a1* transcription in renal proximal tubule cells and kidney. We previously reported cooperation between VDR, the C/EBP family of transcription factors and the SWI/SNF complex in the 1,25(OH)_2_D_3_ mediated regulation of CYP24A1 as well as an inhibitory effect of PRMT 5 methyltransferase (a type II PRMT) on *Cyp24a1* transcription [26]. We now report that CARM1 mediates coactivator activity in the presence of SRC2/GRIP1 (and can also cooperate with G9a) in 1,25(OH)_2_D_3_ induced *Cyp24a1* transcription. In addition, we show that VDR recruitment at *Cyp24a1* VDR binding sites in kidney and MPCT cells in response to 1,25(OH)2D3 is accompanied by dimethylation of histone H3 at arginine 17 and that treatment of MPCT cells by a specific inhibitor of histone 3 arginine 17 methylation results in a significant inhibition of 1,25(OH)_2_D_3_ induced *Cyp24a1* expression, further suggesting a role for CARM1 in VDR-mediated induction of renal CYP24A1. 1,25(OH)_2_D_3_ activated methylation of histone 3 at arginine 17 (a principal site of CARM1 action) at VDR binding sites would result, at least in part, in a modification of histone tails and thus participate together with coregulators that acetylate histones in the regulation of VDR-dependent target gene expression. Our findings also provide evidence that the effect of CARM1 as a VDR transcriptional coactivator is not specific for CYP24A1 and that CARM1 can also act as a repressor of second messenger-mediated induction of CYP27B1 transcription involved in the synthesis of 1,25(OH)_2_D_3_. These findings suggest that cooperativity between p160 coactivators and methyltransferases plays a fundamental role in VDR-mediated transcriptional activity and in the regulation of calcium homeostasis by vitamin D.

## 2. Materials and Methods

### 2.1. Materials

The small molecule inhibitor, ellagic acid dihydrate (TBBD), CARM1 polyclonal antibody (09-818), anti-actin polyclonal antibody (A2066), and salmon calcitonin (05232401) were purchased from Sigma-Aldrich Co. (St. Louis, MO, USA). Sepharose Protein A was obtained from Rockland Immunochemicals, Inc. (Gilbertsville, PA, USA). VDR (C-20) and OPN (P-18) antibodies and anti-rabbit (sc-2004) and anti-goat (sc-2020) horseradish peroxidase-(HRP) conjugated) antibodies were purchased from Santa Cruz Biotechnology, Inc. (Santa Cruz, CA, USA). Asymmetric dimethyl H3R17 antibody (ab8284) was obtained from Abcam (Cambridge, MA, USA). Donkey anti-rabbit IgG HRP (711-035-152) was purchased from Jackson ImmunoResearch Laboratories, Inc. (West Grove, PA, USA). 1,25(OH)_2_D_3_ was purchased from Cayman Chemical Co. (Ann Arbor, MI, USA). Pre-stained protein markers and polyvinylidene difluoride (PVDF) membranes were obtained from Bio-Rad. The luciferase assay system (E1501) was purchased from Promega (Madison, WI, USA). RiboZol RNA Extraction Reagent was purchased from Amresco (Solon, OH, USA).

### 2.2. Cell Culture

Fetal bovine serum (FBS) and charcoal-stripped FBS were obtained from Gemini Biological Products (Calabasas, CA, USA). Cell culture media, 0.05% trypsin-EDTA and penicillin, streptomycin, and neomycin mixture were purchased from Invitrogen. COS-7 African green monkey kidney cells and human colon adenocarcinoma cell line (Caco-2), obtained from the American Type Culture Collection (Manassas, VA, USA), and AOK-B50 porcine renal proximal tubular cells (LLCPK1 cells) that express PTH/PTHrP type I receptors as well as calcitonin receptors [27,28], were cultured in DMEM. Mouse proximal convoluted tubule (MPCT) cells, provided by Peter Friedman [29], were maintained in DMEM plus Ham’s F12 nutrient mixture (DMEM-F12 GlutaMax). MC3T3-E1 mouse osteoblastic cells (Riken Cell Bank, Tsukuba, Japan) were cultured in α-MEM. Mouse embryonic fibroblast cells from CARM1 replete or deficient embryos (13^+/+^ and 20^−/−^) were generously provided by M.T. Bedford and maintained in DMEM. Media were supplemented with 8–10% heat-inactivated FBS and 1% penicillin, streptomycin, and neomycin antibiotic mixture. Cells were grown in a humidified incubator with an atmosphere of 95% air–5% CO_2_ at 37 °C. For treatments, cells were grown to 60–70% confluence and changed to media supplemented with 2% charcoal-dextran-treated FBS before treatment. Cells were treated with vehicle or test compounds for the durations and with the concentrations indicated in Results and figure legends.

### 2.3. Animals

C57BL/6J mice were exposed to a 12 h light, 12 h dark cycle. Food and water were given ad libitum. Animal experimental protocols were reviewed by the Rutgers, New Jersey Medical School Animal Care and Use Committee. For vitamin D administration studies, male C57BL/6 mice 2 months or 12 months of age were fed a 0.8% strontium, 0.02% calcium, vitamin D-deficient diet (Teklad Diet TD 00562; from Envigo, Madison, WI, USA) for 7 days to inhibit endogenous renal synthesis of 1,25(OH)_2_D_3_ [30]. The serum calcium levels of the strontium-fed mice before injection with vehicle or 1,25(OH)_2_D_3_ were less than 7.5 mg/dL. Mice were treated by intraperitoneal injection with either vehicle (9:1 mix of propylene glycol and ethanol; C mice) or a single dose of 1,25(OH)_2_D_3_ (10 ng/g body weight (bw) in a 9:1 mix of propylene glycol and ethanol) for 3 h before termination. The single-dose protocol was chosen because suboptimal induction of *Cyp24a1* in kidney results 3 h after 1,25(OH)_2_D_3_ injection [31]. Kidneys were isolated, snap-frozen in liquid nitrogen, and processed for RNA or ChIP analysis as described below.

### 2.4. Plasmids, Transient Transfections and Luciferase Assay

The luciferase reporter construct of the rat *Cyp24a1* −298/+74 promoter (containing vitamin D response elements at −258/−244 and −151/−137, crucial for maximal 1,25(OH)_2_D_3_ induction [32] and the intronic VDR-binding sites at +35 and +39 [33]), was used for transfection in COS-7 cells and in mouse embryonic fibroblast cells. pSG5.Ha vectors encoding GRIP1(5-1462), GRIP1ΔAD2, CARM1, CARM-E267Q, and G9a were kindly provided by M. Stallcup (University of Southern California Keck School of Medicine, Los Angeles, CA, USA). The hVDR expression plasmid pAVhVDR was obtained from J.W. Pike (University of Wisconsin-Madison, Madison, WI, USA). For transfection studies in AOK-B50 cells, the mouse *Cyp27b1* promoter −1651/+22 was placed upstream of a luciferase reporter gene in the pGL2b vector and provided by H. F. DeLuca (University of Wisconsin-Madison, Madison, WI). pMEX-CCAAT/enhancer-binding protein-β (C/EBPβ) was a gift of Simon Williams (Texas Tech University, Lubbock, TX, USA). Cells were transfected using Lipofectamine 2000 (Invitrogen) and treated as described in Results. Time-course studies with calcitonin indicated a peak of activation at 6 h (1 nM calcitonin resulted in a 20–21 fold induction in transcription using the −1651/+22 promoter construct). Studies were performed at 24 h (suboptimal conditions), consistent with previous studies examining transcriptional effects of calcitonin (100 nM) [34]. After treatment, cells were washed twice with phosphate-buffered saline (PBS) and harvested by incubating with 1x passive lysis buffer, supplied by the dual luciferase reporter assay kit (Promega, Madison, WI, USA). For transfection in COS-7 cells, embryonic fibroblast cells and AOK B-50 cells, cells were seeded in a 24-well culture dish 24 h prior to transfection at 70–80% confluence. Empty vectors were transfected to maintain equal concentrations of total DNA. The luciferase activity assay was performed according to the manufacturer’s protocol and normalized to values of pRL-RSV-Renilla luciferase as an internal control. The luciferase data represent the mean ± SE from 3 transfected cultures and are representative of at least 3 independent transfection experiments.

### 2.5. RT-PCR Analysis and Western Blot

For RT-PCR analysis, total RNA was extracted using Ribozol extraction reagent. Two micrograms of total RNA were reverse transcribed using Superscript III Reverse Transcriptase (Invitrogen) according to the manufacturer’s protocol. *Cyp24a1*, *Trpv6,* glyceraldehyde-3-phosphate dehydrogenase (*Gapdh*), or *β-Actin* levels were assessed by semiquantitative RT-PCR as previously described [26,35]. For each primer set, PCR cycles were chosen so that amplification was conducted in the linear range of amplification efficiency. Primers and annealing temperatures (Ta) were as follows: *Cyp24a1,* forward, 5′-GTGCGGATTTCCTTTGTGAT-3′ and reverse, 5′-ATTGTCTTCGCTAGAGCCCA-3′, Ta = 55 C; *Trpv6,* forward, 5′-ATCGATGGCCCTGCGAACT-3′ and reverse, 5′-CAGAGTAGAGGCCATCTTGTTGCTG-3′, Ta = 57 C; *Gapdh*, forward, 5′-TCACCATCTTCCAGGAGCG-3′ and reverse, 5′-CTGCTTCACCACCTTCTTGA-3′, Ta = 57 C; *β-Actin*, forward 5′-CCTGTGGATCTGACAGCTGAA-3′ and reverse 5′-TCCCAAATCGGTTGGAGATA-3′. Ta = 52 C. The resulting PCR products were subjected to electrophoresis on a 1% agarose gel containing ethidium bromide, and bands were visualized under UV light. Gel data were recorded using the Gene Genius Bio Imaging System (Syngene, Frederick, MD, USA). Data were normalized for the expression of *Gapdh* or *β-Actin* within the sample.

For Western blot analysis, total protein was prepared and analyzed for protein concentration by the Bradford method, and 30 µg of denatured protein from total cell extracts was loaded onto a 4–20% gradient sodium dodecyl sulfate-polyacrylamide gel (Bio-Rad Laboratories, Inc., Hercules, CA, USA), separated by electrophoresis and transferred onto a PVDF membrane for Western blot analysis using an enhancer chemiluminescent detection system (Denville Scientific, Swedesboro, NJ, USA). β-Actin was used as a loading control.

### 2.6. Chromatin Immunoprecipitation (ChIP) Assay

ChIP was performed as described previously [26,36,37]. Briefly, for in vivo ChIP assays, C57BL/6 wild-type male mice 2 months or 12 months of age were treated by intraperitoneal injection with either vehicle or the indicated concentration of 1,25(OH)_2_D_3_/g body weight, as described above. For in vitro ChIP assays, MPCT and MC3T3-E1 cells were cultured to 95% confluence prior to the experiment and then treated with either vehicle or 1,25(OH)_2_D_3_ as indicated in the Results and figure legends. Kidneys or cells were isolated, rinsed with cold phosphate buffered saline and subjected to a cross-linking reaction with 1% formaldehyde for 15 min. For both in vivo and in vitro ChIP assays, the cross-linking reaction was stopped by adding glycine to a final concentration of 0.125 M for 5 min. For in vivo ChIP, after neutralization of the fixative, kidney tissue was homogenized using a Potter–Elvehjem glass Teflon homogenizer, and cells were collected via 70 µm nylon cell strainers (BD Falcon). Cells were then washed with PBS, pelleted and lysed sequentially in 5 volumes of 5 mM Pipes, pH 8.0, 85 mM KCl, 0.5% Nonidet P-40 containing a protease inhibitor cocktail (Roche, Laval, QC, Canada) and then in 3 volumes of 1% SDS, 5 mM EDTA, 50 mM Tris–Cl, pH 8.1, for 20 min. The lysate was sonicated to an average DNA size of 500 bp using a Fisher Model 100 Sonic Dismembranator at a power setting of 2 as assessed by agarose gel electrophoresis. The sonicated extracts were precleared with Sepharose Protein A beads (100 µL, Rockland Immunochemicals), and immunoprecipitations were performed using either a control IgG antibody or the indicated experimental antibody. DNA fragments were purified using Qiagen QIAquick PCR purification kits (Qiagen Inc. Valencia, CA, USA) and subjected to PCR using the primers designed to amplify the fragment containing the proximal murine VDR *Cyp24a1* binding sites at −258 and −151 [upper, 5′-GGT TAT CTC CGG GGT GGA GT-3′; lower: 5′-AGT GGC CAA TGA GCA CGC-3′]. PCR products were resolved in 1% agarose gel and visualized using ethidium bromide staining. PCR was carried out in the linear range of DNA amplification. Analysis of input DNA (5%) was acquired prior to precipitation (input). DNA acquired prior to precipitation was collected and used as the input. DNA acquired from immunoprecipitates performed with IgG was subjected to PCR using the primers designed to amplify the fragment containing the proximal murine VDR *Cyp24a1* binding site in order to exclude non-specific binding.

### 2.7. Statistical Analysis

Results are displayed as mean ± standard errors of the means (SEMs). Data were analyzed using Student’s *t* test or analysis of variance (ANOVA) and additionally with Benjamini and Hochberg in a post hoc test to consider significant differences between groups (*p* < 0.05).

## 3. Results

### 3.1. 1,25(. OH)_2_D_3_ Induction of Cyp24a1 in Renal Proximal Tubule Cells Is Repressed by TBDD, a Specific Inhibitor of CARM1

Although previous studies noted a major role of methylation in transcriptional regulation of steroid hormones and identified CARM1 as a transcriptional secondary coactivator for steroid receptors (primarily by methylating H3R17), very little is known about the role of methyltransferases in VDR-mediated transcription. Since CYP24A1 has an essential role in the control of calcium homeostasis, regulating blood levels of 1,25(OH)_2_D_3_ by its 1,25(OH)_2_D_3_ and 25(OH)D_3_ catabolic activity, we examined a possible requirement of H3R17 methylation for VDR-mediated *Cyp24a1* expression using cells from renal proximal tubule, the site of regulated expression of CYP24A1 as well as CYP27B1 in the kidney and TBBD, a uniquely specific inhibitor of H3R17 methylation and a target for CARM1 methyltransferase activity [38]. Analysis of *Cyp24a1* expression in MPCT cells treated with vehicle (−D), 10^−8^ M 1,25(OH)_2_D_3_ (+D), TBBD alone (25 µM or 100 µM) or 10^−8^ M 1,25(OH)_2_D_3_ in combination with TBBD (25 µM or 100 µM) for 24 h revealed a significant inhibition of 1,25(OH)_2_D_3_ induction of *Cyp24a1* in the presence of increasing concentrations of TBBD (*p* < 0.05 compared to +D alone (Figure 1A)).

### 3.2. Reduction in 1,25(OH)_2_D_3_ Induced Cyp24a1 Transcriptional Activity in CARM1 Deficient Cells

Although inhibition of H3R17methylation resulted in repression of 1,25(OH)_2_D_3_ induction of *Cyp24a1,* further studies were performed to determine a role of CARM1 in VDR *Cyp24a1* gene transcription using mouse embryo fibroblasts from CARM1^−/−^ mice. CARM1 deficient embryos were able to survive during fetal development but died perinatally [39]. Two mouse embryonic fibroblast cell lines, CARM1^+/+^ (13^+/+^) and CARM1^−/−^ (20^−/−^), have been established to facilitate studies of CARM1 function despite neonatal lethality in CARM1deficient mice [39]. Using the rat *Cyp24a1* −298/+74 construct containing vitamin D response elements at −258/−244 and −151/−137, crucial for maximal 1,25(OH)2D3 induction of *Cyp24a1* [32] and the intronic VDR binding sites at +35 and +39 [33], we found, in the presence of 10^−10^ M or 10^−8^ M 1,25(OH)_2_D_3,_
*Cyp24a1* transcriptional activity was reduced 40–50% in CARM1^−/−^ cells compared to CARM1^+/+^ cells, further suggesting a role for CARM1 in 1,25(OH)_2_D_3_ regulation of *Cyp24a1* (Figure 1B).

### 3.3. CARM1 Mediates Coactivator Synergy in the Presence of GRIP1 and Also Cooperates with G9a in 1,25(OH)_2_D_3_ Induced Cyp24a1 Transcription

In order to examine the functionality of CARM1 as a transcriptional coactivator for VDR-mediated Cyp24a1 transcription, we initially used the COS-7 African Green Monkey kidney fibroblast-like cell line in which VDR is absent (as are other steroid receptors) and endogenous levels of cooperating transcription factors or coactivators are low or absent. COS-7 cells were transfected with the *Cyp24a1* −298/+74 construct and VDR expression vector and treated with suboptimal 1,25(OH)_2_D_3_ (10^−10^ M) in the presence or absence of GRIP1 (a primary coactivator that binds to VDR and other steroid receptors) and CARM1 expression vectors. Expression of GRIP1 stimulated transcription 2 fold above the 1,25(OH)_2_D_3_ induced response. However, transfection of GRIP1 in combination with CARM1 resulted in a significant enhancement of *Cyp24a1* transcriptional activity 4–5 fold above the 1,25(OH)_2_D_3_ induced response (*p* < 0.05 compared to activation with GRIP1; maximal 1,25(OH)_2_D_3_ induced transcriptional activity in the presence of GRIP1 and CARM1 was 45 fold). Although G9a, a histone H3 Lys-9 methyltransferase, is generally associated with transcriptional repression, it has also been shown to act as a coactivator for steroid receptors with GRIP1 and to cooperate synergistically with GRIP1 and CARM1 in steroid receptor-mediated transcription [23]. Thus, coactivator synergy between CARM1, GRIP1 and G9a was also examined. Transfection of CARM1 in combination with GRIP1 and G9a resulted in a synergistic enhancement of 1,25(OH)_2_D_3_ induced Cyp24a1 transcriptional activity (*p* < 0.05 compared to each methyltransferase alone +D or compared to activation with GRIP1 +D; maximal 1,25(OH)_2_D_3_ *Cyp24a1* transcriptional activity was 64 fold) (Figure 1C). Thus, CARM1 can function as a coactivator for VDR in the presence of GRIP1 and can also cooperate with G9a in 1,25(OH)_2_D_3_ mediated *Cyp24a1* transcription. To determine if CARM1 methyltransferase activity is required for its coactivator activity in the regulation of *Cyp24a1,* a CARM1 mutant (E267Q) that does not have detectable methyltransferase activity but retains the ability to bind to GRIP was used [17]. Substitution of the CARM1 mutant for wild-type CARM1 reduced coactivator synergy by 50%, indicating the importance of CARM1 enzymatic activity in the coactivator function of CARM1 (Figure 1D). When a GRIP1 mutant that lacks the AD2 domain, which binds to CARM1, was substituted for wild-type GRIP1, reporter gene activity was markedly reduced (Figure 1D). Thus, the AD2 domain of GRIP1 was necessary for the cooperative synergistic activity.

### 3.4. Dimethylation of Histone 3 at Arginine 17, Which Is Mediated by CARM1, Occurs at Cyp24a1 VDR Binding Sites in a 1,25(OH)_2_D_3_ Dependent Manner in MPCT Cells and in Mouse Kidney

Since the CARM1 mutant experiments indicated the importance of methyltransferase activity in the coactivator function of CARM1, and since H3R17 methylation is a target for CARM1, ChIP assays were performed to determine whether VDR recruitment at Cyp24a1 VDR binding sites in MPCT cells and kidney in response to 1,25(OH)_2_D_3_ is accompanied by dimethylation of histone H3 at arginine 17. 1,25(OH)_2_D_3_ treatment of MPCT cells was found to initiate H3R17 methylation at Cyp24a1 VDR binding sites (Figure 2A). In order to determine the specificity for 1,25(OH)_2_D_3_ mediated H3R17 methylation in VDR-mediated CYP24A1 regulation for the PCT, ChIP assays were performed in MC3T3 mouse osteoblastic cells. It has been suggested that CYP24A1 in non-renal VDR target cells may have a role in regulating intracellular levels of 1,25(OH)_2_D_3_. ChIP analysis showed recruitment of VDR, which was accompanied by enhanced H3R17 at VDR binding sites in MC3T3 cells, suggesting that histone methylation at R17 (a principal site of CARM1 action) is not specific for the kidney PCT (Figure 2B).

To examine the in vivo significance of 1,25(OH)_2_D_3_ mediated H3R17 methylation in the regulation of CYP24A1, in vivo ChIP analysis was performed. Kidneys isolated from 2- and 12-month-old C57BL/6 mice were used to examine possible age related alterations in VDR recruitment and in the enhancement of H3R17 methylation in response to 1,25(OH)_2_D_3_. At 2–3 months of age, there is an increased capacity to absorb calcium in response to 1,25(OH)_2_D_3_ that decreases by 12 months of age [40,41]. In the C57BL/6 mouse by 12 months of age, in addition to decreased intestinal calcium absorption, there are significant changes in bone architecture that have been reported to be similar to age-related changes in bone in humans [42,43]. Renal Cyp24a1 has previously been reported by us and others to increase with age, suggesting that increased catabolism of 1,25(OH)_2_D_3_ contributes to the decreased capacity of the adult compared to young animals (as well as humans) to maintain calcium balance and bone mass [44,45]. Analysis of young and adult mouse kidney revealed an age-related increase in renal Cyp24a1 expression as well enhanced H3R17 methylation and enhanced recruitment of VDR at Cyp24a1 VDR binding sites in the kidney of the 12-month-old C57Bl/6 mice, suggesting a mechanism to explain, in part, the age-related increase in renal Cyp24a1 expression (Figure 2C,D)

### 3.5. CARM1 Also Enhances 1,25(OH)_2_D_3_ Induction of Trpv6 and Osteopontin

Studies using Caco-2 cells and MC3T3 cells transfected with CARM1 showed enhanced expression of *Trpv6* and osteopontin (OPN), respectively, suggesting that the effect of CARM1 as a transcriptional coactivator for VDR is not specific for CYP24A1 (Figure 3A,B). Since CARM1 VDR coactivator activity occurs in the presence of GRIP1, the enhancement of these 1,25(OH)_2_D_3_ targets by CARM1 may involve, in part, the presence of endogenous GRIP1 or other SRCs in these cells. Further studies are needed.

### 3.6. CARM1 Acts as a Repressor of Calcitonin Activation of Cyp27b1 Transcription: Reversal by C/EBPβ

CARM1 has previously been shown to act as a dual function co-regulator that can repress or activate gene expression, depending on the context of transcription factors [20,21]. As previously noted, in addition to histones, CARM1 can have non-histone substrates and modify coregulatory proteins in order to modulate gene expression [12,18,19]. Early studies showed that during pregnancy and lactation, calcitonin levels are increased under normocalcemic conditions and correlated to increased serum 1,25(OH)_2_D_3_ levels [46,47]. It had been proposed that calcitonin has a physiological function to increase 1,25(OH)_2_D_3_ levels during pregnancy and lactation when the need for calcium is increased [46,47]. Further studies in our lab showed that calcitonin strongly induces *Cyp27b1* transcription in kidney cells and that the transcription factor C/EBPβ is a mediator of this induction [48]. Since previous studies noted that CARM1 can interact with and methylate C/EBPβ and that methylated C/EBPβ becomes transcriptionally inactive [49], we examined a possible role of CARM1 in the regulation of calcitonin-mediated transcription of *Cyp27b1.* Using AOK-B50 cells (porcine renal proximal tubular cells that express PTH as well as calcitonin receptors [27,28]) transfected with the mouse *Cyp27b1* −1651/+22 construct, treatment with calcitonin resulted in a 7 fold induction in *Cyp27b1* transcription, as previously described [48] (Figure 4). Co-transfection of CARM1 (0.2 µg) in AOK-B50 cells significantly decreased calcitonin-mediated induction of *Cyp27b1* transcription (Figure 4). Co-expression of C/EBPβ (0.05 μg) reversed CARM1-mediated repression of *Cyp27b1* transcription (Figure 4), suggesting cross-talk between CEBPβ and arginine methylation by CARM1 as a regulatory mechanism for renal *Cyp27b1* transcriptional activity.

## 4. Discussion

Vitamin D is a principal factor that maintains calcium homeostasis. In addition, it has been suggested that there are beneficial effects of 1,25(OH)_2_D_3_ that are not involved in calcium regulation, including effects on both innate and adaptive immunity [5,50,51] and inhibition of mammary, colon and skin cancer [52]. Although VDR plays a critical role in sustaining calcium balance, and evidence suggests additional beneficial effects of 1,25(OH)_2_D_3_, we are now only beginning to understand the multistep process requiring a combination of transcriptional coactivators. It has been suggested that VDR coactivators are master regulators of 1,25(OH)_2_D_3_ action and that cell- and gene-specific functions are mediated through differential recruitment by 1,25(OH)_2_D_3_ of coregulatory complexes and finely controlled by post-translational modifications. Understanding coactivator regulation of VDR-mediated gene expression is essential to our understanding of the mechanisms involved in the multiple biological actions of vitamin D. Since coactivators may selectively regulate specific pathways, VDR coactivators may represent a new class of drug targets for therapeutic intervention. Since precise regulation of vitamin D metabolism is critical to the maintenance of calcium homeostasis, in this investigation, we examined the regulation by 1,25(OH)_2_D_3_ of renal CYP24A1. Our findings provide evidence that CARM1 may play a fundamental role as a VDR coactivator in VDR-mediated induction of CYP24A1 and thus in the regulation of 1,25(OH)_2_D_3_ catabolism and the maintenance of calcium homeostasis.

The importance of GRIP1 was noted in our studies. The GRIP1 mutant lacking the binding site for CARM1 (C-terminal AD2 region; ΔAD2) compromised the coactivator synergy of CARM1, G9a, and GRIP1 in VDR-mediated transcription. Early studies noted that CARM1 but not other PRMTs were involved in the coactivator function of G9a [23]. The C terminal activation domain of GRIP1 (AD1) is involved in recruitment of CBP/p300, whereas CARM1 binds to the AD2 domain [23]. CARM1 and p300/CBP are secondary coactivators. A ligand-dependent interaction of GRIP1 with the AF2 domain of steroid receptors (GRIP1 as a primary coactivator) is mediated via the steroid receptor transcription factor interaction domain containing a series of conserved motifs (LXXLL sequence motifs, where L is leucine and X is any amino acid) termed NR box [53]. Thus, GRIP1 contributes to transcription by inducing recruitment of secondary coactivators that can acetylate or methylate histones, other coactivators, transcription factors and non-histone proteins. SRC-1, the first member of the SRC family of coactivators, which binds directly to steroid hormone receptors, was discovered in the laboratory of Dr. B. O’ Malley in 1995 [54], followed by the identification of GRIP1 (SRC-2) [55,56,57,58,59,60,61]. Although in our studies we examined GRIP1 as a primary coactivator, the other members of the SRC family also serve similarly as primary coactivators for steroid receptors to recruit secondary coactivators [62,63]. Recent studies by Moena et al. [64] noted the importance of SRC-1 and histone acetylation in the regulation by 1,25(OH)_2_D_3_ of CYP24A1 in bone cells. Studies by Meyer and Pike using ChIP-seq analysis of 1,25(OH)_2_D_3_ target genes in LS180 human intestinal epithelial cells found that SRC-1 binding was most highly correlated to VDR/RXR binding at enhancer elements and tracked most closely to 1,25(OH)_2_D_3_ regulated genes, including *CYP24A1* and *TRPV6* [65]. SRC-1 was not detectable before treatment with 1,25(OH)_2_D_3_. Morena et al. also noted that CARM1 knockdown prevented H3R17me2a enrichment at CYP24A1 VDR binding sites in response to 1,25(OH)_2_D_3_ and repressed 1,25(OH)_2_D_3_ induction of CYP24A1 mRNA. SRC-1 knockdown prevented the H3R17 enrichment at the VDR binding sites, further suggesting an interrelationship between CARM1-dependent histone methylation, p160 coactivators and histone acetylation [64]. We previously noted that PRMT5, a type II PRMT, by interacting with BRG1 (a component of the SWI/SNF chromatin remodeling complex, that cooperates with VDR in the induction of *Cyp24a1* transcription), is part of a complex involved in the transcriptional repression of *Cyp24a1* [26]. Although we did not address mechanisms involved in the repression of *Cyp24a1* in the absence of 1,25(OH)_2_D_3_, we noted that symmetrical dimethylation of H3R8 and H4R3 was increased in 1,25(OH)_2_D_3_ treated cells transfected with PRMT5 [26]. Our findings suggested that these histone methylations may mediate, in part, the PRMT5 repression of 1,25(OH)_2_D_3_ induced *Cyp24a1* transcription at times when protection against hypercalcemia is not needed. In support of our findings, Morea et al. noted that PRMT5 knockdown in ROS17.28 cells treated with 1,25(OH)_2_D_3_ resulted in a decrease in symmetrical H4R3 at VDR binding sites and an increase in *Cyp24a1* [64]. These findings provide further evidence for a role of methyltransferases in VDR-mediated transcription.

Although our findings indicate a VDR coactivator function for CARM1, it should be noted, however, that the regulation of SR function by coactivators is complex (over 400 coregulators of nuclear receptors that can regulate the assembly of transcription complexes have been identified [66]). It has been reported that most coregulators can function as a coactivator or as a repressor [63]. Although most studies have focused on GRIP1 as a coactivator, it should be noted that GRIP1 can also function in transcriptional repression (for example, in GR-mediated repression of IL-8 [67] and in ER-mediated repression of TNF alpha [68]). As indicated previously, G9a can act as a coactivator corepressor [22]. G9a self-methylation enables HP1γ (heterochromatin 1 γ) to bind to G9a, which facilitates recruitment of RNA, resulting in activation of gene transcription for some GR target genes. Phosphorylation of G9a-like protein (GLP) prevents binding of HP1γ to G9a and reduces the coactivator function of G9a [24]. CARM1 can also function as a transcriptional repressor as well as an activator. It has been reported that CARM1 methylation of CBP/p300 blocks the interaction between CBP/p300 and cAMP response element binding protein (CREB) and thus acts as a repressor in the cAMP signaling pathway [21]. In our studies, we also noted that CARM1 can act as a transcriptional repressor of *Cyp27b1*, which is involved in the synthesis of 1,25(OH)_2_D_3_, supporting the role of CARM1 as a dual-function coregulator. Our previous results noted that calcitonin, which has been proposed to have a role in increasing 1,25(OH)_2_D_3_ levels during pregnancy and lactation [46,47], strongly induces Cyp27b1 mRNA and protein levels in kidney cells [48]. The transcription factor CEBPβ was found to mediate the calcitonin induction of *Cyp27b1* transcription [48]. Previous studies noted that when C/EBPβ binds to and is methylated by CARM1, it displays repressor functions [49]. Thus, CARM1 repression of the induction of *Cyp27b1* transcription by calcitonin may involve methylation of C/EBPβ. Reversal of the repression by overexpressed C/EBPβ points to a delicate balance between the level of C/EBPβ and its posttranslational modification and may be due to alleviation of sequestration of C/EBPβ by CARM1.

Adding to the complexity, coactivators interact in large protein complexes that add posttranslational modifications not only on histones but also on steroid receptors, other proteins in the transcription complex and additional coregulators. Further studies are needed to determine whether VDR is methylated by CARM1 and the effect of possible methylation by CARM1 of additional coregulators recruited by 1,25(OH)_2_D_3_ to *Cyp24a1* and other target genes. In addition, since ablation of each SRC was noted to have a different phenotype and to regulate different physiological processes [66], in vivo studies are also needed to determine the effect of ablation of primary coactivators as well as secondary coactivators on VDR-mediated gene regulation and function.

In this investigation, we examined the role of CARM1 in the regulation of 1,25(OH)_2_D_3_ induced *Cyp24a1* in renal proximal convoluted tubule cells as well as in mouse kidney. Recent studies by Meyer et al. identified kidney-specific enhancer modules localized exclusively in renal proximal tubules that mediate PTH upregulation and FGF23 and 1,25(OH)_2_D_3_ suppression of *Cyp27b1* [69,70]. *Cyp24a1* is transcriptionally regulated in a reciprocal manner by PTH, FGF23 and 1,25(OH)_2_D_3_ [5]. Meyer and Pike also identified enhancers localized only in the kidney that mediate PTH and FGF23 regulation of *Cyp24a1*. Regulation of *Cyp24a1* by 1,25(OH)_2_D_3_ was found to be mediated by regulatory elements present in both kidney and non-renal target cells [71]. Early studies using microdissected nephron segments reported that CYP27B1 and CYP24A1 are exclusively expressed in the renal proximal tubule [11]. However, it has been a matter of debate since the expression of CYP27B1 and CYP24A1 has been identified in other renal cell types as well as non-renal cell types [72,73]. Recent genomic studies by Cusanovich DA et al. [74] using ATAC-seq and single cells (including kidney cells as well as non-renal cells) from C57BL/6 mice revealed the presence of open chromatin across the genome. Pike et al. noted that the regulatory sites with open chromatin were the sites previously defined by Meyer and Pike as enhancers that mediate the regulatory expression (by PTH, FGF23 and 1,25(OH)_2_D_3_) of *Cyp27b1* and were exclusively in the cells of the proximal tubule (not in other cells of the kidney or in non-renal cells), confirming that the proximal tubule is the site of the regulated production of 1,25(OH)_2_D_3_ [75]. The regulatory expression (at least by PTH and FGF23) of *Cyp24a1* was also noted to be proximal tubule-specific [75]. Since both *Cyp27b1* and *Cyp24a1* are in the same cell type in the kidney, they can, therefore, coregulate the endocrine production of 1,25(OH)_2_D_3_. Although the PCT is the site of the regulatory expression of *Cyp27b1* and *Cyp24a1*, the roles of CYP27B1 and CYP24A1 in non-renal target cells still remain to be defined. It has been suggested that CYP24A1 in non-renal cells can limit the levels of 1,25(OH)_2_D_3_ in cells, preventing toxicity, or may modulate 1,25(OH)_2_D_3_ function in target cells. The production of 1,25(OH)_2_D_3_ by alveolar macrophages in patients with granulomatous disorders has been noted, resulting in hypercalcemia in these patients [76]. However, the role of CYP27B1 under normal conditions at extrarenal sites has still been a matter of debate [77]. Further studies are needed in order to understand the physiological role of non-renal production and regulation of 1,25(OH)_2_D_3_.

## 5. Conclusions

In conclusion, our results support the role of CARM1 as a VDR coactivator of *Cyp24a1* transcription in renal proximal tubule cells and kidney. CARM1 was found to mediate coactivator synergy in the presence of GRIP1 (a coactivator with acetyl transferase activity) and also with G9a lysine methyltransferase and GRIP1 in 1,25(OH)_2_D_3_ induction of *Cyp24a1* transcription (Figure 5). TBDD, a specific inhibitor of CARM1, resulted in an inhibition of 1,25(OH)_2_D_3_ induced *Cyp24a1* expression in mouse proximal tubule (MPCT) cells. Methylation of histone 3 at arginine 17, which is mediated by CARM1, occurred at *Cyp24a1* VDR binding sites in response to 1,25(OH)_2_D_3_, further supporting a role for CARM1 in 1,25(OH)_2_D_3_ induction of renal *Cyp24a1* expression. Although CARM1 was found to act as a coactivator of VDR-mediated transcription, CARM1 repressed second messenger-mediated *Cyp27b1* transcription, suggesting modulation by CARM1 of different regulatory pathways involved in the biological function of 1,25(OH)_2_D_3_. Thus, depending on the context of transcription factors, CARM1 may act as a dual-function coregulator within the vitamin D endocrine system to stimulate VDR-mediated transcription or repress second messenger-induced transcription. Understanding the coregulators involved in VDR regulation of target genes involved in maintaining calcium homeostasis and age-related dysregulation may facilitate the development of drugs that act through transcriptional mechanisms to prevent or reverse age-related deterioration of calcium homeostasis.

## Figures and Tables

**Figure 1 cells-12-01407-f001:**
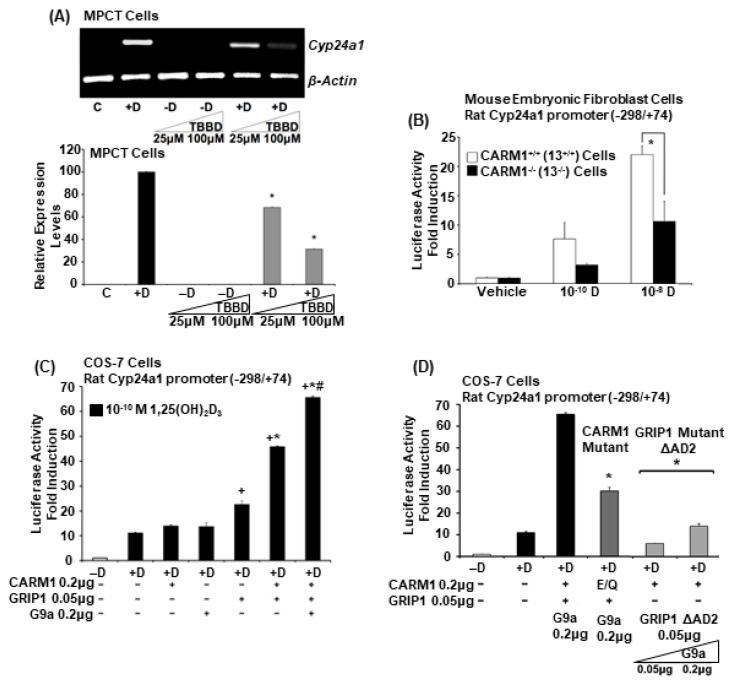
1,25(OH)_2_D_3_ induction of *Cyp24a1* is repressed by a specific inhibitor of CARM1 (TBBD) (**A**) and in CARM1-deficient cells (**B**). CARM1 mediates coactivator synergy in the presence of GRIP1 and cooperates with G9a in 1,25(OH_2_D_3_ induced *Cyp24a1* transcription (**C,D**). (**A**) RT-PCR analysis of *Cyp24a1* in MPCT cells treated with vehicle (−D), 10^−8^ M 1,25(OH)_2_D_3_ (+D), TBBD alone (25 μM or 100 μM), or 10^−8^M 1,25(OH)_2_D_3_ in combination with TBBD (25 μM or 100 μM) for 24 h. Results are reported as the mean ± SE of triplicate cultures. * *p* < 0.05 compared to cells treated with 1,25(OH)_2_D_3_ alone (+D). The data are representative of at least 4 independent experiments. (**B**) *CARM1^+/+^* or *CARM1^−/−^* mouse embryonic fibroblast cells were transfected with the rat *Cyp24a1* promoter luciferase construct (−298/+74, 0.2 μg) and hVDR expression plasmid (0.02 μg). Cells were treated with vehicle, 10^−10^ M 1,25(OH)_2_D_3_, or 10^−8^ M 1,25(OH)_2_D_3_ for 24 h, and luciferase activity was determined. *Cyp24a1* promoter activity is represented as fold induction (mean ± SE) by comparison to basal levels (vehicle-treated). * *p* < 0.05 compared to *CARM1^+/+^* cells treated with 1,25(OH)_2_D_3_. (**C**) COS-7 cells were plated and transfected with the rat *Cyp24a1* promoter construct (−298/+74, 0.2 μg) and hVDR expression plasmid (0.02 μg) in the presence or absence of CARM1 alone (0.2 μg), G9a alone (0.2 μg), GRIP1 alone (0.05 μg) or in combination and treated with vehicle (−D) or 10^−10^ M 1,25(OH)_2_D_3_ (+D) for 24 h. Luciferase activity was determined and represented as fold induction (mean ± SE) by comparison to −D levels. **^+^**
*p* < 0.05 compared to +D alone or +D + CARM1 or +D + G9a; **^*^**
*p* < 0.05 compared to +D in combination with GRIP1; # *p* < 0.05 compared to +D and CARM1 and GRIP1 in combination with G9a. (**D**) COS-7 cells were transfected with the rat *Cyp24a1* promoter construct (−298/+74) and hVDR expression plasmid in the presence or absence of CARM1 wild type or E/Q mutant (0.2 μg), GRIP1 wild type or ΔAD2 GRIP1 mutant (0.05 μg) and G9a (0.05 and 0.2 µg). Cells were treated with vehicle (−D) or 1,25(OH)_2_D_3_ (10^−10^ M). Luciferase activity is represented as fold induction (mean ± SE) by comparison to −D levels. * *p* < 0.05 compared to +D, CARM1, GRIP1 and G9a (0.2 µg) transfected. In separate experiments the CARM1 and GRIP1 mutants also inhibited the enhancement by CARM1 in the presence of GRIP1 (in the absence of G9a) of 1,25(OH)_2_D_3_ induced *Cyp24a1* transcription (data not shown).

**Figure 2 cells-12-01407-f002:**
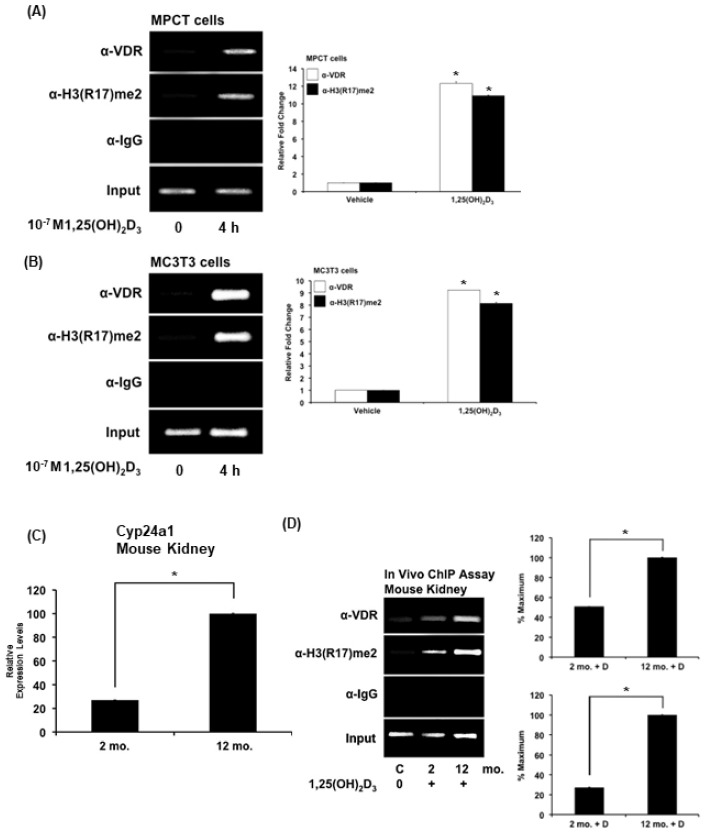
1,25(OH)_2_D_3_ dependent dimethylation of histone H3 at arginine 17 occurs at *Cyp24a1* VDR binding sites in MPCT cells, osteoblastic cells and mouse kidney. For ChIP analysis, cells were treated with vehicle or 10^−7^ M 1,25(OH)_2_D_3_ for 4 h. Cell lysates were subjected to immunoprecipitation with VDR antibody (α-VDR) or dimethyl(R17)H3 antibody (α-H3(R17)me2) or control rabbit IgG. DNA precipitates were isolated and then subjected to PCR using specific primers designed to amplify regions of mouse *Cyp24a1* that contain the VDR binding sites (see Materials and Methods). The bar graphs represent quantification for MDCT cells (**A**) and MC3T3 cells (**B**) of three independent ChIP analyses for each cell type (±SE). * *p* < 0.05 compared to cells treated with vehicle. (**C**) Age-related increase in renal *Cyp24a1* in 12-month-old male mice compared to 2-month-old male mice. Results are reported as relative expression levels ± SE; n = 4–5 mice per age group. (**D**) For in vivo ChIP analysis, vitamin D-deficient male mice were injected with either vehicle (C, control) or 1,25(OH)_2_D_3_ (10 ng/g body weight) for 3 h before termination. Cells were collected from kidney tissue and subjected to immunoprecipitation as described above. DNA precipitates were isolated and subjected to PCR (see Methods and Materials). Enhanced recruitment of VDR and enhanced H3R17 methylation at *Cyp24a1* vitamin D response elements was observed in the kidney of 12 month old mice compared to 2 month old mice. The bar graph represents the mean percentages of the maximal response ±SE of three independent ChIP analyses. *, *p* < 0.05, 12-month-old mice injected with 1,25(OH)_2_D_3_ compared to 2-month-old mice injected with 1,25(OH)_2_D_3_.

**Figure 3 cells-12-01407-f003:**
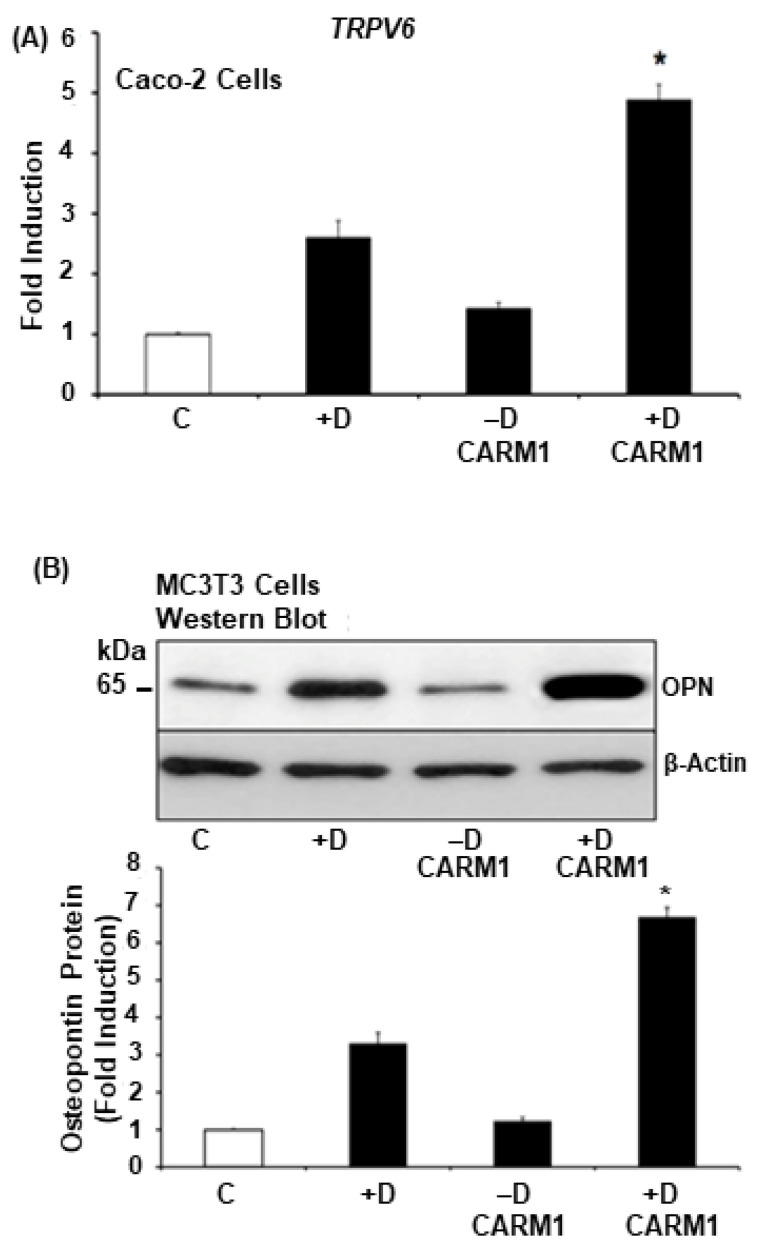
CARM1 also enhances 1,25(OH)_2_D_3_ induction of other vitamin D targets. (**A**) *TRPV6* expression in Caco-2 cells treated with vehicle (−D) or 10^−7^ 1,25(OH)_2_D_3_ (+D, 24 h) in the presence or absence of CARM1 expression vector (1 μg). Results are reported as the mean fold induction ±SE compared to control (open bar); n = 3 separate experiments. *, *p* < 0.05 compared with CARM1 0.2 μg alone or 1,25(OH)_2_D_3_ alone. (**B**) Western blot analysis of osteopontin (OPN) in MC3T3 cells treated with vehicle (−D) or 10^−8^ M 1,25(OH)_2_D_3_ (+D, 24 h) in the presence or absence of CARM1 expression vector (1 μg). The results are reported as the mean fold induction compared to control (open bar) ±SE; n = 3 separate experiments.

**Figure 4 cells-12-01407-f004:**
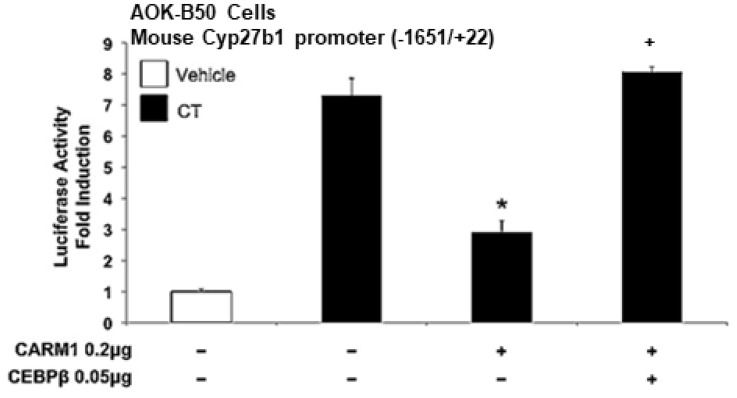
CARM1 acts as a repressor of calcitonin activation of *Cyp27b1* transcription, and C/EBPβ reverses the CARM1-mediated repression. AOK-B50 kidney cells were transfected with 0.3 μg of the mouse *Cyp27b1* promoter construct (−1651/+22) in the presence or absence of CARM1 (0.2 μg) and CEBPβ (0.05 μg). After 24 h, cells were treated with vehicle (open bar) or 100 nM calcitonin (CT; black bar) for another 24 h. *Cyp27b1* promoter activity was measured by luciferase activity, as described in Figure 1, and reported as fold induction (mean ± SE) by comparison to vehicle treated (open bar). Co-expression of CARM1 repressed the transcriptional activation by CT; * *p* < 0.05 compared with CT treated). Co-expression of CEBPβ reversed the CARM1-mediated repression of CT induction (**^+^**, *p* < 0.05 compared with CARM1 0.2 μg + CT). CEBPβ reversal of CARM1-mediated repression was similarly observed using *Cyp27b1* promoter constructs −144/+22 and −85/+22, which retain the C/EBP site and the responsiveness to CT (48) (not shown).

**Figure 5 cells-12-01407-f005:**
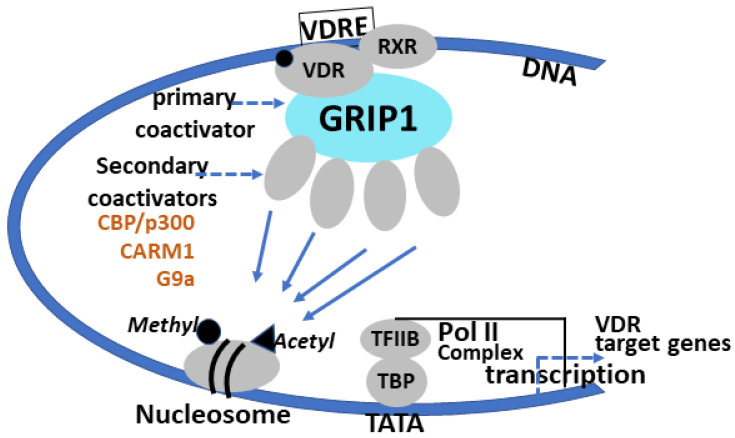
Histone methyltransferases, together with histone acetyltransferases, serve as transcriptional coactivators for VDR. 1,25(OH)_2_D_3_ (black circle) activates VDR, which heterodimerizes with RXR. VDR/RXR interacts with specific enhancer elements or vitamin D response elements (VDREs) in and around target genes. 1,25(OH)_2_D_3_-VDR recruits coregulatory complexes that can include histone methyltransferases (CARM1 and G9a) and histone acetyltransferases (GRIP1,CBP/p300) that add methyl or acetyl groups on histones (shown here) or other coregulators or proteins in the transcription complex, contributing to the transcriptional activation of VDR target genes. (Adapted from Stallcup, M.R. et al. J Steroid Biochem and Mol Biol 85, 2003, 139. M.R. Stallcup was the first to identify CARM1 as a regulator of transcription (ref. [16]).

## Data Availability

All data are presented in the manuscript.

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
