# Peer review of "Role of Coactivator Associated Arginine Methyltransferase 1 (CARM1) in the Regulation of the Biological Function of 1,25-Dihydroxyvitamin D3"

_cells, 2023, doi:10.3390/cells12101407_

Round 1

Reviewer 1 Report

As nicely stated in the introduction, the cellular activities of 1,25-Dihydroxyvitamin D3 (1,25(OH)2D3), the hormonally active form of vitamin D ac-tivates the nuclear vitamin D receptor (VDR) to mediate the transcription of target genes involved in calcium homeostasis as well as in non-classical 1,25(OH)2D3 actions. The pathways and intercritical activation and stop points are still being elucidated.  In this study CARM1, an arginine methyltransferase, was found to mediate coactivator synergy in the presence of GRIP1 (a primary coactivator) and to cooperate with G9a, a lysine methyltransferase, in 1,25(OH)2D3 induced transcription of Cyp24a1 (the gene involved in the metabolic inactivation of 1,25(OH)2D3). In mouse proximal renal tubule (MPCT) cells and in mouse kidney chromatin immunoprecipitation analysis demon-strated that dimethylation of histone H3 at arginine 17, which is mediated by CARM1, occurs at Cyp24a1 vitamin D response elements in a 1,25(OH)2D3 dependent manner. Treatment with TBBD, an inhibitor of CARM1, repressed 1,25(OH)2D3 induced Cyp24a1 expression in MPCT cells, further suggesting that CARM1 is a significant coactivator of 1,25(OH)2D3 induction of renal Cyp24a1 expression. CARM1 was found to act as a re-pressor of second messenger mediated induction of the transcription of CYP27B1 (in-volved in the synthesis of 1,25(OH)2D3),.supporting the role of CARM1 as a dual func-tion coregulator. The  findings supported by strong methodology  indicate a key role for CARM1 methyltransferase in the regulation of the biological function of 1,25(OH)2D3.

The only comments from this reviwer are

1. The paper would benefit from a figure that demonstrates the pathways studied and how it then translates to a clinical endpoint If possible.

2. A few sentences in the conclusion on how this finding increases our understanding and could translate to the clinical actions of vitamin D

Reviewer 2 Report

This manuscript is interesting, However, some modifications have to be performed prior to acceptance.

1)      Since this work is complicated, a graphical abstract is required to better understand and absorb more audience.

2)      Interaction between Vit. D3 and H3R17 methylation is not clear. Moreover, the relationship between extracellular Ca2+ (perhaps secretion) and cARM1-mediated H3R17 methylation has not been described clearly. These are having to be presented by figure.

3)       Resolution of figure 1 is not good enough. E.g. Y-aces cannot be read. 

Reviewer 3 Report

1) Authors should provide clear figures in the revised manuscript. All the figures are pixelated, which makes it difficult to understand the results.

2) Figure captions should be shortened. Include detailed explanation of these figures in the results/discussions section and not in the captions.

3) In the introduction, you need to connect the state of the art to your paper goals. Please follow the literature review by a clear and concise state of the art analysis. This should clearly show the knowledge gaps identified and link them to your paper goals. Please reason both the novelty and the relevance of your paper goals.

4) The work is well written and provides good results, but their discussion can be deepened. Also, compare your results with the results of other researchers.

5) Better discuss industrial aspects of your research results in details. Some discussions are necessary for the introduction to provide the readers with a big picture.

6) Conclusions must go deeper, it would be more interesting if the authors focus more on the significance of their findings regarding the importance of the interrelationship between the obtained results and the journal scope in the sector context, and the barriers to do it, what would be the consequences, in the real world, in changing the observed situation, what would be the ways, in the real world, to change/improve the observed situation.

7) Several grammatical errors were spotted, please correct the same. Also, authors are advised to strictly follow the journal’s formatting guidelines.

Round 2

Reviewer 2 Report

1) The authors misunderstood about graphical abstract. A graphical abstract is a figure to describe the whole of the work as graphic images. They could search on the web and find samples. They thought we need an image inside the abstract.  It is required for a better understanding of the purpose of the work. 

2) In Figure 1, the legend of X - Y is so small. The authors said they had 400 DPI. Itis regarding the resolution. I am talking about font size. When anyone read the text in 100% magnification, it should be readable. 

Author Response

1) with regard to the graphical abstract, yes I misunderstood.  However Fig. 5 included in the revision now describes the whole of the work as a graphical image.  

2) thank you for indicating that the font in Fig. 1 needs to be increased.  The font has now been increased for Fig. 1 x and y axes. 

Reviewer 3 Report

The authors have satisfactorily addressed all my comments and concerns.

Author Response

as indicated by reviewer 3 we have satisfactorily answered all concerns